# Synthetic Polypeptides with Cationic Arginine Moieties Showing High Antimicrobial Activity in Similar Mineral Environments to Blood Plasma

**DOI:** 10.3390/polym14091868

**Published:** 2022-05-02

**Authors:** Kuen Hee Eom, Shuwei Li, Eun Gyeong Lee, Jae Ho Kim, Jung Rae Kim, Il Kim

**Affiliations:** 1School of Chemical Engineering, Pusan National University, Busandaehag-ro 63-2, Geumjeong-gu, Busan 46241, Korea; reimreim@pusan.ac.kr (K.H.E.); lishuwei0325@pusan.ac.kr (S.L.); rud6063@pusan.ac.kr (E.G.L.); j.kim@pusan.ac.kr (J.R.K.); 2Department of Physiology, School of Medicine, Pusan National University, Busandaehak-ro, Mulgeum-eup, Yangsan-si 50612, Korea; jhkimst@pusan.ac.kr

**Keywords:** amino acids, antimicrobial peptides, arginine, *N*-carboxyanhydrides, polypeptides, ring-opening polymerization

## Abstract

Translocation of cell-penetrating peptides is promoted by incorporated arginine or other guanidinium groups. However, relatively little research has considered the role of these functional groups on antimicrobial peptide activity. A series of cationic linear-, star- and multi-branched-poly(*L*-arginine-*co*-*L*-phenylalanine) have been synthesized via the ring-opening copolymerizations of corresponding *N*-carboxyanhydride monomers followed by further modifications using the *N*-heterocyclic carbene organocatalyst. All the polymers are characterized by the random coiled microstructure. Antibacterial efficacy, tested by the gram-positive *B. subtilis* bacteria and the gram-negative *E. coli* bacteria, was sensitive to the structure and relative composition of the copolymer and increased in the order of linear- < star- < multi-branched structure. The multi-branched-p[(*L*-arginine)_23_-*co*-(*L*-phenylalanine)_7_]_8_ polymer showed the best antibacterial property with the lowest minimum inhibitory concentration values of 48 μg mL^−1^ for *E. coli* and 32 μg mL^−1^ for *B. subtilis*. The efficacy was prominent for *B. subtilis* due to the anionic nature of its membrane. All of the resultant arginine moiety-containing polypeptides showed excellent blood compatibility. The antibiotic effect of the copolymers with arginine moieties was retained even in the environment bearing Ca^2+^, Mg^2+^, and Na^+^ ions similar to blood plasma. The cationic arginine-bearing copolypeptides were also effective for the sterilization of naturally occurring sources of water such as lakes, seas, rain, and sewage, showing a promising range of applicability.

## 1. Introduction

Antibiotics are used to prevent and treat bacterial infections. Antimicrobial resistance (AMR) is caused by mutations in bacteria due to the use of antibiotics; this is a significant concern because the resistant subject is bacteria, not humans or animals. Antibiotic-resistant bacteria are more difficult to treat than non-resistant bacteria, leading to increased medical costs, extended hospitalization, and increased mortality [1]. Infectious diseases that can be treated with general treatment courses of antibiotics have become more difficult to treat due to the spread of AMR. The World Health Organization states that the emergence of antibiotic-resistant bacteria is causing the world to once again enter a post-antibiotic era in which humans can die from common infections [2]. There is an urgent need for the research and development of new antibiotics.

Antimicrobial peptides (AMPs) are emerging as alternatives to traditional antibiotics, and AMPs such as nisin, gramicidin, polymyxins, daptomycin, vancomycin, and melittin have been used [3,4]. Existing antibiotic mechanisms interfere with cell wall synthesis, protein synthesis, or DNA replication [5] and can stimulate the immune system or interact with bacterial membranes to kill bacteria [6]. Among them, the mechanism of interaction with the bacterial membrane selectively binds to the negatively charged bacterial cell membrane with cationic amino acids like lysine, histidine and arginine, and hydrophobic amino acids like valine, alanine, phenylalanine, and leucine [7].

AMPs are attracting attention as next-generation antibiotics as biodegradable polymers that do not remain in the environment and are difficult for bacteria to develop resistance to because they primarily have mechanisms that interact with bacterial cell membranes. AMP is obtained only from gene-modified microorganisms; therefore, the production rate is low and the cost is high [8].

To circumvent this problem, AMP-mimetic polypeptides (AMPPs), which reproduce AMP-like structures with synthetic polymers, are emerging as alternatives [9]. Among them, the ring-opening polymerization (ROP) of α-amino acid N-carboxyanhydrides (NCAs) can synthesize polypeptides with cationic amino acids and hydrophobic amino acids on a large scale [10]. Therefore, AMPPs synthesized by the ring-opening polymerization of NCAs have the advantage of low cost. Because of lysine NCA’s storability, well-understood mechanism, and high yield, many AMPPs utilizing lysine have been reported because lysine is the easiest to synthesize of the cationic amino acids [11,12,13]. AMPPs with arginine moieties have been reported less because they are relatively difficult to synthesize [14,15,16,17,18,19,20,21]. In addition, the synthesis of AMPs with arginine is important in antibiotic studies. AMPs with arginine units have been reported to have stronger membrane interactions and perturbation properties than AMPs with lysine counterparts [22,23,24,25,26,27,28].

Lysine has been reported to have a significant decrease in antibiotic efficacy due to the presence of divalent cations, but arginine has been reported to have a relatively low antibiotic effect [29,30]. There are divalent cations such as Ca^2+^ and Mg^2+^ in biological matrices, which means that the antibiotic effect may decrease when AMP enters the body. Thus, we have studied AMPPs that were less disturbed by divalent cations and maintained antibacterial properties by synthesizing polypeptides with arginine moieties as positive charges.

In our previous study, we reported an imidazolium hydrogen carbonate (IHC) catalyst that synthesizes topological nanoengineered antimicrobial polypeptides of various structures through the ROP of NCAs [10]. IHC-mediated NCA polymerizations have many advantages, including metal-free procedure, absence of any toxic impurities, and a living pathway [10]. It was also possible to synthesize polypeptides of various structures by only modifying the initiator. Here, several proportions of linear- (*l*-), star- (*s*-), and multi-branched- (*mb*-) poly(*L*-lysine-*co*-*L*-phenylalanine) [p(Lys-*co*-Phe)] have been synthesized and the *L*-lysine units were further modified to *L*-arginine units to yield poly(*L*-arginine-*co*-poly(*L*-phenylalanine) [p(Arg-*co*-Phe)] counterparts. According to the antibacterial tests made for the gram-positive *B. subtilis* bacteria and the gram-negative *E. coli* bacteria, the efficacy was sensitive to the to the structure and compositions of the copolymer and increased in the order *l*- < *s*- < *mb*-structure. All of the p(Arg-*co*-Phe) copolymers showed excellent blood compatibility. Notably, the antibiotic effect was retained in the presence of Ca^2+^, Mg^2+^, and Na^+^ ions and the copolymers show their efficiency for the sterilization of naturally occurring sources of water such as lakes, seas, rain, and sewage.

## 2. Materials and Methods

Anhydrous *N*,*N*-dimethylformamide (DMF), trifluoroacetic acid (TFA), triphosgene, trimethylamine (TEA), KHCO_3_, calcium chloride, magnesium chloride, and solvents like tetrahydrofuran (THF), diethyl ether, hexane, and methanol were purchased from Fisher Scientific Korea (Incheon, Korea) and used without purification. *L*-phenylalanine (Phe), 1,3-Diisopropylimidazolium chloride, and ε-benzyloxycarbonyl-*L*-lysine (Cbz-Lys) were purchased from Fisher Scientific Korea and stored under N_2_. Hexylamine (HA); tris-(2-aminoethyl)amine (TREN); polyethylenimine, branched (PEI; number average molecular weight = 600); 1*H*-pyrazole-1-carboxamide monohydrochloride; and 33 wt% hydrogen bromide solution in acetic acid were purchased from Merck Korea (Seoul, Korea) and stored under nitrogen. All reactions and polymerization were carried out under nitrogen atmosphere using the Schlenk techniques. *Escherichia coli BL21* (*E. coli*) and *Bacillus subtilis ATCC 6633* (*B. subtilis*) were obtained from the Korean Collection for Type Cultures (Jeollabuk-do, Korea). Sewage was provided by the East Busan Sewage Treatment Plant (Busan, Korea).

### 2.1. Synthesis

The 1,3-diisopropylimidazolium hydrogen carbonate ([*^i^*PrNHC(H)]^+^[HCO_3_]^−^) catalyst precursor was synthesized by reacting 1,3-diisopropylimidazolium chloride with KHCO_3_ following a reported procedure [10].

Poly(ε-carbobenzoxy-*L*-lysine), p(Cbz-Lys), and p(Cbz-Lys-*co*-Phe) with different arms were synthesized using HA, TREN, and PEI as initiators according to methods previously described [10]. Figure 1 illustrates general synthetic procedures. For the deprotection, 5 mmol of p(Cbz-Lys) and p(Cbz-Lys-*co*-Phe) were dissolved in TFA in a round-bottom flask equipped with glass stoppers and stirrer bars. Two equivalents of HBr (33% *v*/*v*) per carboxyl group were added and the mixture was stirred for 4 h at 25 °C. The solution was then poured into an excess amount of cold diethyl ether and purified in a dialysis bag (1 kDa) for 2 d. After freeze-drying, white solid p(Lys) and p(Lys-*co*-Phe) were obtained.

The guanidination of δ-NH_2_ was achieved with 1*H*-pyrazole-1-carboxamide monohydrochloride as a guanidylating agent at pH 9.5 in aqueous medium [14,15,16,17]. In a typical experiment, p(Lys-*co*-Phe) was dissolved in DMF in a Schlenk flask under nitrogen. PCH (1.2 times the number of lysine units) and TEA (the same number of moles as PCH) were dissolved in DMF, added to a syringe, administered dropwise to the polypeptide solution, and reacted for 3 d. This solution was then purified in a dialysis bag (1000 Da) for 4 d and lyophilized. Finally, p(Arg-*co*-Phe) was obtained in a white solid state and used for the antibacterial activity assay.

### 2.2. Antibacterial Test

A growth inhibition assay was performed to determine the antibacterial effects of p(Lys-*co*-Phe) and p(Arg-*co*-Phe) with different structures. The model bacteria *E. coli* (a gram-negative bacillus) and *B. subtilis* (a gram-positive, catalase-positive bacterium) cells were cultured overnight in Luria-Bertani (LB) medium at 37 °C. Sterilized LB broth (50 mL) was added to a sterilized culture dish. Sterilized LB broth was added to a bacterial broth and 3.6 mL of the broth containing the bacterial suspension diluted at a concentration of 5 × 10^7^ CFU mL^−1^ was added to 5 mL of sterile quartz cuvette. The standard solution of p(Lys-*co*-Phe) or p(Arg-*co*-Phe) was prepared in distilled water at a concentration of 5.12 mg mL^−1^, and then the standard solutions were diluted in series. A mixture of bacteria and copolymer was obtained by adding a standard stock solution (400 μL) to each quartz cuvette. 400 μL of distilled water was added to 3.6 mL of a diluted stock containing bacteria suspended in solution to use as a control group. The bacteria in the cuvette were cultured overnight in a rocking bed at 80 rpm and 37 °C. In the suspended fluid, the optical density (OD) value of the bacteria was monitored at a wavelength of 600 nm every 3 h using UV-vis spectroscopy. The minimum inhibitory concentration (MIC) value was defined as the lowest concentration of an antibiotic at which bacterial growth is completely inhibited. All analyses were performed three times at different dates.

Plate counting was performed to determine whether antibacterial performance was observed in natural water gathered from different sources such as lake, rain, sewage, and sea. The copolymer standard solution was prepared in distilled water at a concentration of 5.12 mg mL^−1^, and the standard solution was diluted in series. Natural water and copolymer solution were added in a ratio of 9:1 to create an experimental group sample; natural water and distilled water were added as controls in a ratio of 9:1. The samples were diluted to different concentration gradients and 10 μL of the diluted samples were spread evenly on solid LB agar medium at 30 °C overnight for colony growth and observation.

### 2.3. Hemolysis Test

The hemolytic activity of p(Lys-*co*-Phe) and p(Arg-*co*-Phe) was investigated in the range between 100 μg mL^−1^ and 3000 μg mL^−1^. Blood from mice was washed three times with phosphate buffered saline (PBS) and then diluted to 10% (*v*/*v*) with PBS. Blood (90 μL) was first added to a 1.5 mL microfuge tube. Then, 10 μL of each standard stock solution of copolymer was added. Blood suspensions were also added to blank PBS as the negative control. PBS containing 0.2% Triton-X was used as a positive control. After incubation at 37 °C for 1 h, each mixture was centrifuged at 5000 rpm for 5 min. An aliquot of 70 μL of the supernatant was transferred to each well of a 96-well plate and the OD value was read at 350 nm. The hemolysis ratio relative to the TX-100 control group was calculated as [(*A_sample_* − *A_negative_*)/(*A_sample_* − *A_positive_*)] × 100%, where *A_sample_*, *A_negative_*, and *A_positive_* are the OD values of the supernatant from the incubated sample, negative control, and positive control, respectively (Appendix A).

### 2.4. Characterization

The ^1^H and ^13^C nuclear magnetic resonance (NMR) spectra were recorded on a Varian Unity 400 MHz spectrometer; shifts were reported from tetramethylsilane in a downfield of 1 millionth and referenced residual solvent peaks. Fourier transform infrared (FT-IR) spectral data were collected for film samples cast on KBr disks, which were measured using a Shimadzu IRPrestige-21 spectrophotometer (Shimadzu, Kyoto, Japan) with 32 scans per experiment at a resolution of 1 cm^−1^. UV-vis spectrometer analysis was performed at a scan rate of 300 nm·min^−1^ using a Shimadzu UV-1650 PC (Shimadzu, Kyoto, Japan). The absorbance and transmittance spectra of the copolymer/bacteria solution were recorded at 600 nm. Circular dichroism (CD) analysis was performed using a Jasco J-1500 spectrometer (Jasco, Easton, MD, USA) with 1 cm quartz cells at 25 °C. The copolymers were dissolved in PBS to 0.5 mg mL^−1^. Wavelengths between 190 nm and 260 nm were analyzed; the integration time was 1 s and the wavelength step was 0.2 nm. Distilled water was used as a reference solvent and five scans were recorded for all the copolymers. The secondary structure of copolymers was analyzed using the CD Multivariate Calibration Creation Program in Spectra Manager^TM^ Version 2 (Jasco, Easton, MD, USA).

The zeta potential was measured using a Malvern Zetasizer Nano ZS device (Malvern Pananalytical, Malvern, UK), equipped with a monochromatic coherent He-Ne laser (633 nm) as the light source and a detector that detects light scattered at an angle of 173° and a constant temperature of 25 °C. Zeta potential was measured at a concentration of 1 mg mL^−1^ in PBS. All samples were filtered through a 0.45 μm nylon filter before measurement and performed in triplicate. Scanning electron microscopy (SEM) was used to detect the morphology of the pathogen on a JCM-5700 Scanning Electron Microscope (JEOL USA, Peabody, MA, USA). Bacteria were treated with the copolymer at a 3 × MIC value and cultured for 4 h in a shaking bed at 37 °C, and the control group was prepared under the same conditions without adding copolymer. Both the treated bacterial group and the control group were washed three times with PBS, and the samples were fixed with 2.5% (*v*/*v*) glutaraldehyde in phosphate buffer (0.1 M) overnight at 4 °C. The fixed bacterial suspension was washed three times with PBS to remove excess fixers and dehydrated in an ethanol series (30%, 50%, 70%, 80%, 95%, and 100% in PBS (0.01 M)—100% was repeated three times). The solution was dried by moving it to a cover slide in a 100% ethanol suspension state. Finally, the samples were ready for SEM analysis.

## 3. Results and Discussion

### 3.1. Synthesis and Characterization of Copolypeptides

For the synthesis of p(Arg-*co*-Phe) with different topologies, a series of ROPs of Cbz-Lys NCA and Phe NCA were performed using three different amine initiators—HA, TREN, and PEI that result in *l*-, *s*-, and *mb*-copolymers, respectively—in the presence of an organocatalyst, [*^i^*PrNHC(H)]^+^[HCO_3_]^−^ (Figure 1). The [*^i^*PrNHC(H)]^+^[HCO_3_]^−^-mediated ROP of amino acid NCA permitted the achievement of rapid and efficient synthesis of well-defined polypeptides and provided control over the polypeptide architecture simply by tuning the type of amine initiators due to its living nature [10]. Table 1 shows the results of polymerizations performed by controlling the initial [monomer]/[initiator]/[catalyst] ratio to 120:1:0.2, 90:1:0.2, and 60:1:0.2, respectively, by varying the relative Cbz-Lys NCA/Phe NCA ratio. According to the molecular weight of the obtained p(Cbz-Lys) and p(Cbz-Lys-*co*-Phe) polymers, the targeted homo- and copolymers were successfully achieved within 20 min of polymerization at 25 °C. As the incorporation of Phe units increases, the solubility of resultant copolymers sharply decreases. However, the deprotected counterparts, p(Lys) and p(Lys-*co*-Phe), were soluble in common NMR solvents. ^1^H NMR spectra of *l*-, *s*-, and *mb*-polypeptides are in Appendix A.

The transformation of ornithine into arginine to prepare arginine-containing peptides by guanylation has long been recognized [21,31]. Our preliminary studies using commercially available guanylating reagents cyanamide, *O*-methylisourea hydrogen sulfate [32], 2-ethyl-2-thiopseudourea hydrobromide [33], and 3,5-dimethylpyrazole-l-carboxamidine nitrate [34] showed insufficient reactivity for practical use. We found that 1*H*-pyrazole-1-carboxamide monohydrochloride was the most efficient and chemically specific guanylation of sterically unhindered δ-NH_2_ (Figure 1).

Figure 2a–c shows ^1^H NMR spectra of *s*-[p(Cbz-Lys)_21_]_3_ (Entry 7 in Table 1), *s*-[p(Lys)_21_]_3_, and its guanylated counterpart (*s*-[p(Arg)_21_]_3_), respectively, and Figure 2d–f shows ^1^H NMR spectra of *l*-[p(Cbz-Lys)_23_-*co*-p(Phe)_7_] (Entry 10), *l*-p[(Lys)_23_-*co*-(Phe)_7_], and *l*-p[(Arg)_23_-*co*-(Phe)_7_], respectively. The methyne group of Cbz-Lys (−NH–C**H**–; c) and methylene proton (−C**H**_2_–NH_2_; g) of the of Cbz-Lys side chain in *s*-p(Cbz-Lys)_21_ are detected at 3.73 ppm and 3.08 ppm, respectively; they appeared at 4.23 ppm and 2.87 ppm, respectively, in *s*-p(Lys)_21_; and 4.25 ppm and 3,07 ppm, respectively, in *s*-p(Lys)_21_. The methylene protons (−C**H**_2_–; d, e, and f) of the other side chain appeared as a multiplet in between 1.18 and 2.05 ppm for *s*-p(Cbz-Lys)_21_, between 1.23 and 1.76 ppm for *s*-p(Lys)_21_, and between 1.20 and 1.75 ppm for *s*-p(Arg)_21_. It is not surprising to observe similar results from the ^1^H NMR spectra of *l*-p[(Cbz-Lys)_23_-*co*-(Phe)_7_], *l*-p[(Lys)_23_-*co*-(Phe)_7_], and *l*-p[(Arg)_23_-*co*-(Phe)_7_] except the appearance of the peaks corresponding to Phe fragments. A similar peak and shift pattern was reported during the transformation of poly(Lys-*co*-Val) to poly(Arg-*co*-Val) [31]. Successful transformations of Cbz-Lys to Lys and to Arg units were also confirmed by the FTIR spectra of *s*-p(Lys)_21_, *s*-p(Arg)_21_, *l*-p[(Lys)_23_-*co*-(Phe)_7_], and *l*-p[(Arg)_23_-*co*-(Phe)_7_] by observing a characteristic stretching band at 1542 cm^−1^ (Appendix A) for the guanidine functionality.

The secondary structures of *s*-p(Arg)_21_ and *l*-p[(Arg)_23_-*co*-(Phe)_7_] were analyzed by CD spectroscopy in PBS at 25 °C. The spectra of both samples had a strong negative band near 200 nm and a few positive shoulder bands above 220 nm (Figure 3), indicating that both samples consist of similar secondary structures. Each absorption band gives rise to different characteristic bands that can be deconvoluted to estimate the secondary structure components of the polymers using the CD Multivariate Calibration Creation Program in Spectra Manager^TM^ Version 2. Through curve-fitting procedures based on a set of reference spectra with known secondary structure components, the four secondary structure components (α-helical, β-sheet, turn, and random coil) could be estimated (see inset in Figure 3). Both *s*-p(Arg)_21_ and *l*-p[(Arg)_23_-*co*-(Phe)_7_] consist of predominantly random coil structures with small portions of β-sheet and turn structures. Both samples show no absorption bands indicating α-helical structures.

### 3.2. Antimicrobial Activities

Prior to performing antimicrobial activities, the potential stability of the p(Arg) and p(Arg-*co*-Phe) colloidal systems was measured at a concentration of 1 mg mL^−1^ in PBS using zeta potential measurement. As summarized in Table 1, the zeta potentials of all polymer particles are more positive than +30 mV, demonstrating their intrinsic stability in buffer solution.

The antibacterial activity of all guanylated polymers including *s*-p(Arg), *s*-p[(Arg)-*co*-(Phe)], *l*-p[(Arg)-*co*-(Phe)], and *mb*-p[(Arg)-*co*-(Phe)] was evaluated by measuring MICs using *B. subtilis* as the gram-positive bacteria and *E. coli* as the gram-negative bacteria (Appendix A). In the antibacterial ability evaluation conducted on LB broth, bacterial proliferation was clearly observed in the control group containing only LB broth and bacteria, but bacteria did not proliferate in the experimental group containing LB broth, bacteria, and polymer. As summarized in Table 1, comparing three *s*-(Arg) polymers (Entry 1, 4, and 7 in Table 1) with different numbers of Arg units, the *s*-(Arg) polymers with the largest number of Arg units shows the lowest MIC value. Comparing *s*-p[(Arg)-*co*-(Phe)] copolymers with different numbers and compositions of each repeat unit, the MIC value tend to decrease as the repeat unit increases and the relative composition of Phe units decreases, demonstrating that the Arg unit is a vital component in identifying antibacterial activity. However, *s*-p[(Arg)_24_-*co*-(Phe)_7_]_3_ polymer shows the lowest MIC value of 48 μg mL^−1^ in *E. coli* and 48 μg mL^−1^ in *B. subtilis*, indicating that the hydrophobic portion affected antibacterial activity to some extent. This polymer shows better antibacterial activity than its linear counterpart, *l*-p[(Arg)_23_-*co*-(Phe)_7_]. The PEI-initiated multi-branching copolymer with similar repeat units in each arm (*mb*-p[(Arg)_23_-*co*-(Phe)_7_]_8_; Entry 11) shows the lowest MIC values: 48 μg mL^−1^ in *E. coli* and 32 μg mL^−1^ in *B. subtilis*. Therefore, the topology of polymers also affects antibacterial efficacy because the topology is important when developing antibacterial agents with membrane destruction mechanisms. This multi-branching copolypeptide also showed the low hemolysis of 32.4% at 3000 μg·mL^−1^, which is 93.75 times higher than its MIC in *E. coli*, demonstrating acceptable blood suitability (Appendix A).

Blood plasma consists of about 90% water and transports nutrients, wastes, antibodies, ions, hormones, etc. Even though ions make up only about 1% by weight of blood plasma, they are the major contributors to plasma molarity, since their molecular weights are much less than those of proteins. NaCl constitutes more than 65% of the plasma ions. Bicarbonate, potassium, calcium, phosphate, sulfate, and magnesium are other plasma ions. Antibacterial effects were tested on *s*-[p(Lys)_23_-*co*-p(Phe)_7_]_3_ (Entry 5, Table 1) and *s*-[p(Arg)_23_-*co*-p(Phe)_7_]_3_ polymers in the absence or presence of additional Ca^2+^ and Mg^2+^ (Figure 4). LB broth contains nutrients and ions necessary for microbial growth, which reduces the likelihood of antibiotic capacity being hindered by nutrients and ions in performance. Additionally, it has lower concentrations of Ca^2+^ and Mg^2+^ ions than plasma. Therefore, the concentrations of Ca^2+^ and Mg^2+^ ions were set to 2 mM, considering the ion concentration in the blood plasma is 2.5 mM for Ca^2+^ and 1.5 mM for Mg^2+^. Both polymers showed high antibiotic efficacy at 64 ug/mL in pure water with a higher efficiency for *s*-[p(Arg)_23_-*co*-p(Phe)_7_]_3_. Specifically, polymer shows a high efficacy for *B. subtilis*. The efficacy decreased in the media bearing Ca^2+^ and Mg^2+^ ions. The decrease in efficacy was apparent for *s*-[p(Lys)_23_-*co*-p(Phe)_7_]_3_, most probably because the amine groups in Lys units may serve as anchors for metal ion chelation with amine groups.

Considering the percentage of salt in blood is about 0.9 percent by weight, antibacterial activity was tested in 1.0% NaCl solution. As illustrated in Figure 4, both *s*-[p(Lys)_23_-*co*-p(Phe)_7_]_3_ and *s*-[p(Arg)_23_-*co*-p(Phe)_7_]_3_ polymers showed better antibacterial activity in salt solution than in pure water. *s*-[p(Arg)_23_-*co*-p(Phe)_7_]_3_ polymer showed better efficacy than *s*-[p(Lys)_23_-*co*-p(Phe)_7_]_3_ polymer and no bacterial proliferation was apparently observed for *B. subtilis*. The antibacterial efficacy of *s*-[p(Arg)_23_-*co*-p(Phe)_7_]_3_ polymer was retained in the presence of additional cations, Ca^2+^ or Mg^2+^. It is worth noting that *s*-[p(Arg)_23_-*co*-p(Phe)_7_]_3_ polymer showed much better activity than *s*-[p(Lys)_23_-*co*-p(Phe)_7_]_3_ polymer in all media. These results indicate that the copolypetides bearing cationic Arg side chains show better antibiotic efficacy than those bearing neutral Lys side chains. The cationic Arg moieties would more easily bind to the negatively charged bacterial cell membrane with relatively lower interference of the various metal cations existing in the media, which is favorable in vivo applications. The low antimicrobial activity in *E. coli* may be due to the fact that a strong interaction between the Lys or Arg moieties and the phospholipid head group prevents translocation of the peptide into the inner leaflet of the membrane [35]. In addition, the hydrophobic Phe residues govern the extent to which the water-soluble p(Arg)-*co*-p(Phe) and p(Lys)-*co*-p(Phe) polymers will be able to partition into the membrane lipid bilayer, and excessive levels of Phe unit can lead to cell toxicity and loss of antimicrobial selectivity [36].

In all cases, p(Arg)-*co*-p(Phe) showed lower efficacy for the growth inhibition of *E. coli* than for the growth inhibition of *B. subtilis*. These results can be inferred from the membrane compositions of both bacterial cells. *E. coli* accumulates three major membrane phospholipids: zwitterionic phosphatidylethanolamine (PE; ~75% of membrane lipids), the anionic lipid phosphatidylglycerol (PG; ~20%), and cardiolipin [37]. The *B. subtilis* lipidome comprises 70% PG, 12% PE, 5% phosphoglycolipid, 4% cardiolipin, 4% diglycosyldiacylglycerol, 2% monoglycosyl diacylglycerol, 2% aminoacyl phosphatidylglycerol, and other components [38]. The cationic Arg moieties in p(Arg)-*co*-p(Phe) strongly interact with anionic PG while displaying repulsive action with zwitterionic PE, as illustrated in Figure 5. Accordingly, p(Arg)-*co*-p(Phe) can be adsorbed onto the *B. subtilis* membrane more strongly than onto the *E. coli* membrane.

Mechanistically, copolypeptides interact with the cytoplasmic membrane first and then accumulate intracellularly, blocking critical cellular processes. Peptide-based antimicrobial action has been studied extensively since it was discovered [6]. The mechanism of action can be divided into two major classes: direct killing and immune modulation. As illustrated in Figure 5, p(Arg)-*co*-p(Phe) accumulates at the surface and self-assembles on the bacterial cell membrane after the initial electrostatic and hydrophobic interactions that are dependent on the composition of the membranes. The direct killing action can be observed by using SEM images (Figure 6). Even though it is difficult to propose the mechanism of action with only the data collected here, the polymer is expected to bind to the surface of the membrane. The membrane-bound peptides then orientate themselves so that the cationic Arg residues face the polar lipid headgroups and the hydrophobic Phe residues face toward the lipid tails. Once a threshold concentration of bound peptide is reached, the hydrophobic Phe residues will penetrate the lipophilic phospholipid layer, eventually destroying the membrane and leading to a formation wrinkle surface tangled with other destroyed bacterial cell membranes.

To evaluate the practical utilization, antibacterial capacity was determined at the *s*-[p(Arg)_23_-*co*-p(Phe)_7_]_3_ concentration of MICx3 in natural waters (Figure 7). The results of the colony formation assays clearly show that *s*-[p(Arg)_23_-*co*-p(Phe)_7_]_3_ is highly effective for the inhibition of bacterial growth in various natural water sources, except rainwater. This material shows particularly high antibacterial ability for seawater bearing ~3.5% NaCl. This result is in line with the previously obtained results that the antibacterial activity of *s*-[p(Arg)_23_-*co*-p(Phe)_7_]_3_ was retained even in the presence of various metal cations, which may broaden the scope of application of the materials of this study. 

## 4. Conclusions

The *l*-, *s*-, and *mb*-p[(Cbz-Lys)-*co*-(Phe)] copolypeptides were synthesized via the random ring-opening copolymerization of corresponding NCAs using organic [*^i^*PrNHC(H)]^+^[HCO_3_]^−^ as a catalyst. The Cbz-Lys units were deprotected to Lys units and further modified to cationic Arg units to yield p[(Arg)-*co*-(Phe)] counterparts. The living character of the copolymerization made it possible to synthesize various copolypeptides with targeted MWs and topologies having predominantly random coil microstructure. The p[(Arg)-*co*-(Phe)] copolymers were intrinsically stable in buffer solution and showed acceptable blood compatibility.

The antibacterial activity of all the guanylated polymers including *s*-p(Arg), *s*-p[(Arg)-*co*-(Phe)], *l*-p[(Arg)-*co*-(Phe)], and *mb*-p[(Arg)-*co*-(Phe)] evaluated by measuring MICs using *B. subtilis* and *E. coli* showed that the Arg unit was a vital component to show antibacterial activity. The *s*-p[(Arg)_24_-*co*-(Phe)_7_]_3_ polymer shows the lowest MIC values of 48 μg mL^−1^ in *E. coli* and 48 μg mL^−1^ in *B. subtilis*. The multi-branching *mb*-p[(Arg)_23_-*co*-(Phe)_7_]_8_ copolymer showed the lowest MIC values: 48 μg mL^−1^ in *E. coli* and 32 μg mL^−1^ in *B. subtilis*.

*s*-[p(Arg)_23_-*co*-p(Phe)_7_]_3_ copolymer was much more effective for inhibiting *B. subtilis* rather than *E. coli* due to the differences of interaction capability between polymer and each bacterial membrane; because it did not lose its antibacterial activity even in the presence of various metal cations, Ca^2+^ or Mg^2+^, and/or Na^+^; and because it showed better activity than its Lys counterpart, *s*-[p(Lys)_23_-*co*-p(Phe)_7_]_3_. *s*-[p(Arg)_23_-*co*-p(Phe)_7_]_3_ was also effective to treat natural water sources such as lake, sea, rain, and sewage. Antimicrobial copolypeptides are unique materials that have shown great promise in treating bacteria while causing them to develop no or only low resistance.

## Data Availability

The data presented in this study are available on request from the corresponding author.

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
