# Peer review of "Synthetic Polypeptides with Cationic Arginine Moieties Showing High Antimicrobial Activity in Similar Mineral Environments to Blood Plasma"

_polymers, 2022, doi:10.3390/polym14091868_

Round 1

Reviewer 1 Report

This paper reports cationic arginine moieties-containing polypeptides showing high antimicrobial activity at similar mineral environments to blood plasma. The antibacterial efficacy for subtilis bacteria and E. coli bacteria was sensitive to the structure and relative composition of the copolymer, and increased in the order linear- < star- < multi-branched structure. The antibiotic effect of the copolymers with arginine moieties was retained even in the environment bearing Ca2+, Mg2+, and Na+ ions similar to blood plasma. The result and discussion are interesting. However, the points in the manuscript still need some improvement. The specific comments are as follows:

  1. The molecular weight of p(Lys-co-Phe) and p(Arg-co-Phe) should be determined by GPC or other method.
  2. The OD value was read at 350 nm, the authors should give the reason. The hemoglobin has the maximum absorption peak at about 540 nm.
  3. The deviation should be added in Figure 4 and 7.
  4. The authors should measure the hydrodynamic size of p(Lys-co-Phe) and p(Arg-co-Phe) by DLS.
  5. Some natural peptides such as daptomycin (introduction International Journal of Pharmaceutics 2022, 615, 121489) and vancomycin have been widely used in antibiotics, which should be included in the introduction.

Reviewer 2 Report

It is suggested to change the article title and use the better one. 

Add more quantitative data to the abstract. 

Add mechanism to the antibacterial activity.

Reviewer 3 Report

The article "Cationic arginine moieties-containing polypeptides showing high antimicrobial activity at similar mineral environments to blood plasma" is devoted to the study of the antimicrobial activity of synthesized polymers. The resulting compounds interact with the cell membrane, allowing them to be used in the fight against bacteria resistant to classical antibiotics.

 I recommend it for publication after the following points are addressed.

1. There are no data on 13C NMR although the method is specified in the experimental part. It is advisable to add spectra to Supporting Information.

2. The method for determining the molecular weight of the synthesized polymers by NMR.

3. What is the polydispersity of the obtained polymers?

4. Why was the GPS/SEC method not used to evaluate molecular mass characteristics? It's better to use this method.

5. It is indicated that poly(L-arginine-co-L-phenylalanine) decomposes in the natural environment and is therefore safe for the environment, but it is not indicated how and by what processes decomposition occurs.

6. The paper does not discuss the differences between Gram-positive and Gram-negative microorganisms. According to the presented data, for some polymers, the differences in MIC are significant. There is no comparison of the obtained data on antimicrobial activity with known/similar antimicrobial polymers.

7. It is advisable to change the color in Fig. 4. Control and poly(L-arginine-co-L-phenylalanine) data are difficult to distinguish due to color similarity.
